# Examining the Relationship between Adaptive Behavior and Intelligence

**DOI:** 10.3390/bs13030252

**Published:** 2023-03-13

**Authors:** Marc J. Tassé, Minje Kim

**Affiliations:** Nisonger Center and Department of Psychology, The Ohio State University, Columbus, OH 43210, USA

**Keywords:** intellectual disability, adaptive behavior, intelligence, intellectual functioning, Diagnostic Adaptive Behavior Scale

## Abstract

Adaptive behavior and intelligence are both essential components of defining and diagnosing intellectual disability. The exact relationship between these two constructs still warrants some clarification. Previous studies have examined the correlation between adaptive behavior and intelligence and have reported differing results. Overall, there seems to be agreement that a modest to moderate correlation exists between adaptive behavior and intelligence and that the strength of this relationship may increase as ability level decreases further below the population mean. Using the Diagnostic Adaptive Behavior Scale and a sample of 57 youth aged from 4 to 21 years old, we examined the correlation coefficients between the full-scale IQ scores and their scores obtained on conceptual, social, and practical adaptive skill domain scores, as well as the total adaptive behavior score. The results obtained indicated a modest to moderate correlation between adaptive behavior and intelligence. The strongest statistically significant correlation coefficient was between the full-scale IQ score and the conceptual adaptive skills domain score (*r* = 0.64). The correlation between the full-scale IQ score and the practical adaptive skills domain (*r* = 0.39) and social adaptive skills domain (*r* = 0.28; ns) were more modest. The correlation coefficient between the full-scale IQ score and the total adaptive behavior score also showed a moderate relationship with intelligence (*r* = 0.46). These findings are consistent with previous research, documenting that adaptive behavior and intelligence are two related but independent constructs. We discuss these findings and their implications.

## 1. Introduction

Intellectual disability is defined by the presence of these three criteria: significant limitations in both intellectual functioning and adaptive behavior (including conceptual, social, and practical skills), and the onset of these limitations during the developmental period [1,2]. Adaptive behavior has been an essential criterion in making a diagnosis of intellectual disability for at least the past 50 years [2,3]. While intelligence is a useful indicator of one’s cognitive assets, it does not accurately reflect how well an individual is functioning in everyday life to meet societal demands and expectations, nor does it solely predict the supports the person may need to function optimally.

Adaptive behavior is defined as the collection of conceptual, social, and practical skills that are learned and performed to meet society’s expectations and demands [2]. Adaptive behavior consists of skills that continue to be learned and used throughout one’s lifetime and, being a distinct construct from intelligence, cannot be inferred from a person’s level of intellectual functioning [4]. A person’s level of adaptive behavior is an indicator of how well an individual typically functions in everyday life, which is also highly predictive of positive life outcomes and has important implications for intervention [5,6].

The American Psychiatric Association (APA) made a significant change in its 5th edition of the Diagnostic and Statistical Manual for Mental Disorders (DSM-5) by substituting adaptive functioning level in place of the individual’s full-scale IQ (FSIQ) score as the construct on which clinicians should rely in guiding their determination of the severity of intellectual disability [1,7]. The rationale put forth by the APA [1,7] was that an individual’s adaptive behavior as a construct better represented the person’s overall level of support needed than the person’s level of intellectual functioning. The exact relationship between adaptive behavior and intelligence is not as well defined as one might expect, considering we have been studying these two constructs for several decades. Adding to this confusion was a sentence in the DSM-5 [1] that stated the following: “To meet diagnostic criteria for intellectual disability, the deficits in adaptive functioning must be directly related to the intellectual impairments described in Criterion A” (p. 28). As stated by Tassé et al. [8], though there might be evidence of a correlational relationship between these two constructs, there is no evidence of any causal relationship. Furthermore, the correlational relationship between these two critical constructs remains modest at best [9,10,11,12]. Several have argued that the two constructs, though moderately correlated, represent two distinct and separate constructs [3,10,13,14]. The APA acknowledged this error in its text revision of the DSM-5 (see DSM-5-TR) [7] by striking the aforementioned sentence from the section related to the diagnostic criteria of intellectual disability.

Studies examining the correlation between the adaptive behavior and intelligence of individuals with intellectual disabilities have reported varying results [15]. Platt et al. [14] reported the correlation coefficients between the Vineland Adaptive Behavior Scale (VABS) and Stanford–Binet IV and Wechsler Intelligence Scale for Children-Revised (WISC-R) on a sample of 99 children (5–19 years old) with intellectual disabilities. Their sample was largely in the mild to moderate range of intellectual functioning deficits (mean FSIQ was 59 (SD = 9) on SB-IV and 61 (SD = 10) on the WISC-R). They reported mean VABS Composite scores of 50 (SD = 9), indicating a group that had, on average, moderate levels of adaptive behavior deficits. Platt et al. reported moderately low correlation coefficients that ranged from *r* = 0.37 to 0.39 between the FSIQ scores and the VABS composite scores for their sample. In another study, Keith et al. [13] examined the correlation coefficients between subscale IQ scores on the Kaufman Assessment Battery for Children (K-ABC) and specific domain scores of the VABS. They reported finding the strongest correlation coefficients between the VABS Communication domain and subscales of the K-ABC (e.g., *r* = 0.43 verbal reasoning; *r* = 0.29 non-verbal reasoning; *r* = 0.36 verbal memory), followed by the VABS Social Skills domain (e.g., *r* = 0.18 verbal reasoning; *r* = 0.16 non-verbal reasoning; *r* = 0.17 verbal memory), and Daily Living Skills domain (e.g., *r* = 0.15 verbal reasoning; *r* = 0.11 non-verbal reasoning; *r* = 0.17 verbal memory). Their findings supported the notion that adaptive behavior and intelligence are separate and distinct constructs [13]. The Vineland Adaptive Behavior Scale, second edition (Vineland-II) user’s manual reported correlation coefficients between adaptive behavior and full-scale intelligence scores, as measured on the Wechsler Intelligence Scale for Children, third edition (WISC-III), as follows: communication *r* = 0.36, daily living skills *r* = 0.25, socialization *r* = −0.39, and composite score *r* = 0.12 [16].

One study conducted with the Adaptive Behavior Assessment Scale, second edition (ABAS-II) reported that children with different neurodevelopmental and socio-emotional disorders (e.g., intellectual disability, communication disorder, Attention-Deficit/Hyperactivity Disorder, and emotional disturbance) showed a small to moderate correlation between ABAS-II scores and full-scale intelligence scores, as measured on the WISC-III: conceptual domain *r* = 0.43, social domain *r* = 0.36, practical domain *r* = 0.38, and composite score *r* = 0.40 [17]. Another study conducted with individuals with intellectual disabilities reported that the correlation between adaptive behavior as measured on the ABAS-II and intelligence as measured on the Wechsler Intelligence Scale for Children, fourth edition (WISC-IV) reported a significant moderate correlation across all domains of adaptive behavior: conceptual domain *r* = 0.48, social domain *r* = 0.31, and practical domain: *r* = 0.32 [18].

McGrew [19] provided an interesting analysis of the correlational relationship between these two constructs by examining the data from a combined 60 correlational analyses extracted from a number of published studies and adaptive behavior scale test manuals. McGrew reported a range in correlational coefficients from *r* = 0.12 to 0.90 with a mean coefficient of *r* = 0.51 (standard deviation *r* = 0.20) and a median correlation coefficient of *r* = 0.48, concluding to an overall moderate correlation between adaptive behavior and intelligence [19].

In a recently published meta-analysis of all available studies, Alexander and Reynolds [9] computed the correlation between adaptive behavior and intelligence, based on a total of 148 independent samples that included a total of 16,464 individuals. Alexander and Reynolds reported an estimated mean correlation coefficient of *r* = 0.51 between adaptive behavior composite standard scores and FSIQ scores. Although their meta-analysis included studies reporting correlations on a number of different adaptive behavior scales, Alexander and Reynolds reported that the composite adaptive behavior scores from the ABAS, Scales of Independent Behavior, and VABS correlated equally with intelligence. This is an important finding since these three instruments are among the handful that are considered as preferred adaptive behavior scales to use when making a determination of intellectual disability [20]. Although Alexander and Reynold’s study provided a robust analysis based on more than 16,000 participants, it does not provide specific information on the correlation between intelligence and the three distinct adaptive behavior domains (e.g., conceptual, social, and practical skills). The American Association on Intellectual and Developmental Disabilities [2], the DSM-5-TR [7], and the World Health Organization [21] in its 11th edition of the International Classification of Diseases, all agree on defining the diagnostic criteria for intellectual disability as requiring the presence of significant deficits in adaptive behavior. All three systems further defined “deficits in adaptive behavior” not based on a composite score but rather on the presence of deficits in one or more of these three adaptive behavior domains (conceptual, social, and practical adaptive skills).

Empirical evidence has consistently supported the assertion that the correlation between adaptive behavior and intelligence varies across the ability level, with the strongest correlation coefficients being at the lower end of the ability scale and more importantly, reporting lower correlation coefficients at or near the decision node (i.e., approximately two standard deviations below the population mean [9,11,22].

It is well established that adaptive functioning is defined as behavior that is learned and performed in relation to the demands of one’s community and these are indexed on chronological age [2,3,7]. Hence, it is crucial to revise our measures of adaptive behavior to keep up with evolving demands and increased complexities of our societies, technological advances being one such example [23]. Thus, it is important to constantly reassess the relationship between the intelligence construct and adaptive behavior construct, especially with the development of newer measures, such as the Diagnostic Adaptive Behavior Scale [24]. It is critical to know the correlation between these two important constructs of adaptive behavior and intelligence because their relationship will impact the probability of a diagnosis of intellectual disability and ultimately its prevalence [9]. The purpose of the present study was to investigate the relationship between adaptive behavior and intelligence based on adaptive behavior scores obtained on a newer published standardized scale of adaptive behavior, the Diagnostic Adaptive Behavior Scale [24], using a sample of children and young adults with and without intellectual disability.

## 2. Methods

### 2.1. Participants

A total of 100 participants were assessed on the Diagnostic Adaptive Behavior Scale (DABS). Among them, 61 participants had also been assessed on an individually administered standardized comprehensive intelligence test. Of this group of 61 participants, four participants were eliminated from the sample because the dates of evaluation separating the assessment on the DABS and the test of intelligence were more than three years (see Figure 1).

The final sample consisted of 57 participants whose chronological age ranged from 4 to 21 years, with a mean age of 12.4 years and a standard deviation of 5.0 years. The distribution of the n = 57 participants across three age groups of the DABS is as follows: 4–8 years old = 29.8%; 9–15 years old = 40.4%; and 16–21 years old = 29.8%. Seventy percent (n = 40) of the assessed individuals were males. The race and ethnic distribution of the sample was as follows: Caucasian = 66.7% (n = 38), African American = 12.3% (n = 7), Asian = 1.8% (n = 1), American Indian/Alaska Native = 1.8% (n = 1), bi-racial or others = 15.8% (n = 9); and Hispanic = 1.8% (n = 1). Fifty-one percent of participants (n = 29) had a FSIQ that was significantly (i.e., <approximately 70) below the population mean and 44% had a diagnosis of autism spectrum disorder.

### 2.2. Procedure

All participants were seen through a university-based interdisciplinary assessment clinic or a child development evaluation clinic at a children’s hospital. Referral reasons included developmental or behavioral concerns and/or purpose of making a determination of intellectual disability or autism spectrum disorder. Because of the nature of the sample, we anticipate that the range of obtained standard scores have a restricted range. It is known that when the range of scores is restricted, a correlation coefficient will likely be reduced [25]. Therefore, to estimate the Pearson correlation coefficients between our variables, we applied Thorndike’s case 2 correction formula [26], which has been shown to provide an adequate correction for restricted ranges [27].

These analyses were conducted as secondary data analyses of clinic records of individuals who had been assessed on the DABS and who also had a prior intellectual functioning assessment on file that had been conducted within three years of the DABS administration. We obtained appropriate review from The Ohio State University Institutional Review Board, which granted us an exemption because we conducted secondary data analyses using a de-identified dataset.

### 2.3. Measures

#### 2.3.1. Intelligence

Only participants who had an intellectual disability were included. Intelligence data from the following assessment instruments were included in this study: Stanford–Binet scales (SB-IV, SB-V), Wechsler’s scale (WAIS-IV, WISC -IV, WIPPSI-R), and Differential Ability Scale (DAS-II).

#### 2.3.2. Adaptive Behavior

Participants’ adaptive behavior was measured using the DABS. The DABS was developed using the item response theory method to include items reflecting conceptual, social, and practical skills, which are three essential components of adaptive behavior [23]. The DABS provides a comprehensive standardized assessment of adaptive behavior. It was developed for use with individuals from 4 to 21 years old to provide precise diagnostic information around the cutoff point where an individual is deemed to have “significant limitations” in adaptive behavior. The DABS yields adaptive behavior results including an overall standard score as well as standard scores for the following three domains: Conceptual skills (literacy; self-direction; and concepts of number, money, and time); Social skills (interpersonal skills, social responsibility, self-esteem, gullibility, naïveté, social problem solving, following rules, obeying laws, and avoiding being victimized); and Practical skills (activities of daily living, self-care, occupational skills, use of money, safety, health care, travel/transportation, schedules/routines). The DABS has reportedly good psychometric properties [23,28,29].

Data were analyzed using SPSS 24.0 [30]. Pearson’s correlation test was conducted to examine the relationship between participants’ DABS standard scores and FSIQ scores.

## 3. Results

We present here the participants’ obtained mean scores on the standardized intelligence test and adaptive behavior scale, as seen in Table 1.

It is true that larger sample sizes generally represent the characteristics of a population more robustly by increasing the statistical power. Although we have a smaller sample size of n = 57, the authors of [31] cited a rule of thumb indicating that a sample size > 50 is a reasonable sample when estimating correlation coefficients to measure the relationship between variables. Pearson correlation tests were conducted to examine the relationship between participants’ FSIQ scores and their DABS standard scores across conceptual, social, and practical skills as well as the DABS Total score. The results of these corrected Pearson correlation coefficients are presented in Table 2, and revealed that participants’ FSIQ was correlated statistically significantly to their conceptual (*r* = 0.64, *p* < 0.001) and practical skills (*r* = 0.39, *p* < 0.05), but the correlation estimate between their FSIQ and social skills (*r* = 0.28, *p* = 0.083; ns) did not achieve statistical significance. The correlation between the FSIQ and the DABS Total score was also statistically significant (*r* = 0.46, *p* = 0.004).

## 4. Discussion

Our study contributes to furthering our knowledge base about the important constructs of adaptive behavior and intelligence and provides new evidence supporting our understanding of this relationship by using data from a new and recently published standardized adaptive behavior scale (i.e., DABS). Our results confirm that the constructs of adaptive behavior and intellectual functioning correlate moderately when comparing the FSIQ with the DABS Total score/overall level of adaptive behavior. Because all our diagnostic systems (i.e., AAIDD, DSM-5-TR, ICD-11) all use functioning at the domain level of adaptive behavior as the criterion for determining intellectual disability, it is important to further examine the relationship between intelligence and the three domains of adaptive behavior. Perhaps not surprisingly, the DABS conceptual domain, which consists of skills related to language, reading/writing, money, time, and number concepts [23], was the domain that correlated the highest with the FSIQ score, even higher than the correlation between FSIQ and the DABS Total score. The other two DABS domains yielded lower correlation coefficients with the participants’ FSIQ. Neither the intellectual functioning nor the adaptive behavior construct can adequately predict the ability level of the other. Our finding of this order of strength of relationship between intelligence and the three domains of adaptive behavior is consistent with previously published studies [18,32]. Our results also reaffirm the notion that the constructs of adaptive behavior and intelligence, although moderately related, remain two separate and independent constructs [11,13,17,19,32,33]. These results confirm the importance of assessing both adaptive behavior and intellectual functioning when establishing a diagnosis of intellectual disability [1,2,7,8,21].

There was a phrase that was introduced in the fifth edition of the Diagnostic and Statistical Manual for Mental Disorders (i.e., DSM-5) that asserted “To meet diagnostic criteria for intellectual disability, the deficits in adaptive functioning must be directly related to the intellectual impairments described in Criterion A” [1] (p. 38). This phrase inadvertently imposed a condition that made a diagnosis of intellectual disability dependent on establishing the presence of a causal relationship linking the deficits in these two independent constructs. All published studies reporting on the correlation between the adaptive behavior scales have consistently indicated a moderate to modest relationship between intelligence and adaptive behavior. Hence, the deficits in adaptive behavior may well be explained by any number of other factors associated with constructs other than the person’s intellectual functioning [9,11]. Likely recognizing this error, the American Psychiatric Association deleted this phrase when they published the text revision of the DSM-5 [7].

It should be noted that although our sample consisted of a large proportion of individuals with ASD (44%), this characteristic of our sample, according to the authors of [9], should not have impacted our correlation coefficients because the relationship between these two constructs does not differ between these two groups.

There are a number of limitations to our study methodology that need some discussion. First, there was variability in the test source of the FSIQ scores used in our analyses. We included a number of different standardized intelligence tests rather than limiting the source of information to only one or two IQ tests. We did limit our source of intelligence results to tests that shared common features with respect to comprehensive and individually administered tests of intelligence that had been administered within a three-year window of the DABS administration. Because our sample did not include individuals with severe to profound deficits in intellectual functioning, we were unable to verify if individuals with the most severe levels of deficits presented higher correlation coefficients than those without significant intellectual functioning deficits or only mild deficits of intellectual functioning. Another limitation that should be considered when interpreting our findings is the small sample size. Although a general rule of thumb cited indicated that a sample > 50 could be sufficient when using correlation coefficients to estimate the relationship between two variables, our small sample size limited our ability to further explore the potential moderating effects of age or gender on our obtained correlation coefficients.

Our findings using the DABS are consistent with previously published findings that reported a moderate to modest correlation between adaptive behavior and intelligence, concluding that these two constructs of human functioning are separate and independent. Our findings are also consistent with previously published research examining the strength of the relationship between intelligence and the separate adaptive behavior domains, finding that the strongest relationship was between conceptual adaptive skills and intelligence, followed by practical adaptive skills, and social adaptive skills.

The AAIDD manual [2] had stated the assumption that with appropriate personalized supports, the life functioning of persons with intellectual disability generally will improve. This is an important notion that too often goes ignored. Previous research has shown that adaptive behavior skills are more strongly related to more favorable life outcomes and personal autonomy in adulthood than the construct of intelligence [33]. Adaptive skills can be taught throughout life and improved adaptive skills will contribute to improving the person’s overall functioning and adaptation across community settings, making it an important outcome to focus on [8]. Another study [34] examined the contribution of parent expectations in relation to levels of adaptive functioning in predicting post-high school outcomes. Parent expectations alone were insufficient to predict better post-school outcomes, concluding that adaptive behavior played a critical role in post-school success for young adults with intellectual disability [34].

Finally, from a diagnostic process approach, these findings support the notion that these two constructs are related but distinct, and thus are equally important when making a determination of intellectual disability. As previously argued in [35], these two constructs are independent and must both be assessed for the purpose of ruling in or ruling out of a diagnosis of intellectual disability. The order of which a construct is assessed first is not relevant as long as both are assessed with equal rigor. Clinicians and researchers cannot assume that the presence of significant deficits in one construct automatically predicts the presence of significant deficits in the other construct, hence both must be independently and rigorously assessed in all cases when making a diagnosis of intellectual disability [35].

## Figures and Tables

**Figure 1 behavsci-13-00252-f001:**
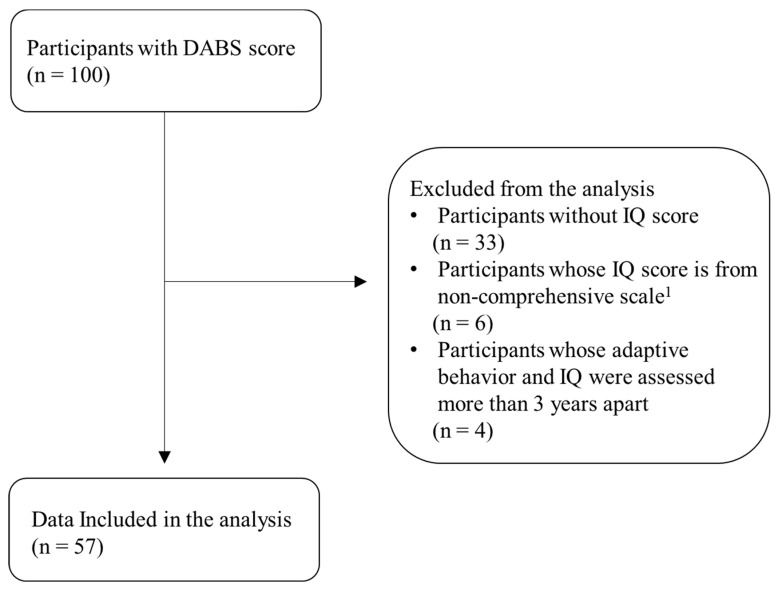
Flow Chart of Data Inclusion. Participants were assessed solely on a developmental scale, abbreviated/short forms or narrow-band test of intelligence (e.g., Test of Non-verbal Intelligence, Comprehensive-Test of Non-verbal Intelligence, Bayley, etc.) were excluded from the analyses.

**Table 1 behavsci-13-00252-t001:** Mean scores and standard deviations on the tests of adaptive behavior and intelligence.

Construct	Mean (SD)
Adaptive Behavior ^1^	
Conceptual skills	72.4 (10.8)
Social skills	75.9 (12.1)
Practical skills	74.8 (10.2)
Total score	70.6 (11.6)
Intelligence ^2^	
Full Scale IQ	71.0 (14.7)

^1^ Adaptive behavior as measured on the Diagnostic Adaptive Behavior Scale (DABS). ^2^ Intelligence as measured on the Wechsler Intelligence Scales, Stanford–Binet Scales, and Differential Ability Scale.

**Table 2 behavsci-13-00252-t002:** Correlation coefficients between intelligence and adaptive behavior.

	Intelligence
Adaptive Behavior	Full scale IQ
Conceptual skills	0.64 **
Social skills	0.39 *
Practical skills	0.28
Total score	0.46 *

** = Correlation is significant at the <0.001 level (2-tailed). * = Correlation is significant at the <0.05 level (2-tailed).

## Data Availability

The data presented in this study are openly available in Science Data Bank at [doi]: 10.57760/sciencedb.07705.

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
