# Peer review of "Examining the Relationship between Adaptive Behavior and Intelligence"

_behavsci, 2023, doi:10.3390/bs13030252_

Round 1

Author Response

Manuscript ID: behavsci-2233109
Type of manuscript: Article
Title: Examining the Relationship Between Adaptive Behavior and Intelligence
Authors: Marc J. Tassé *, Minje Kim
Reviewer: 1

Thank you for your review and comments on our aforementioned manuscript.  We are excited at the possibility of publishing our manuscript in Behavioral Sciences.  We have reviewed your comments and suggested edits to our manuscript and we have made edits to our manuscript to address your comments. We believe these edits, resulting from your review and comments, have strengthened our manuscript. Below is a listing of our responses to Reviewer 1 comments.

  1. It is true that the authors have carried out a descriptive analysis of the standard variables, but one of the variables is defining a very wide physical-psychological variability: the age variable. The authors should make clear the percentage of participants by age range. It is clear that the degree of "maturity" has an impact on intelligence.

Response: We calculated the distribution of participants across the three age groups of the DABS: 4-8 years old; 9-15 years old; and 16-21 years old. These additional descriptive statistics have been included in the “Participants” section of the manuscript.

  1. Regarding the "Results" section, the shortest section of the article, even shorter than the "summary". The little they do is fine, but given the data collection, it would have been interesting to obtain results or to see if there are differences in the results by age range or gender. This would certainly add interest to the study conducted.

Response: We did not further analyze our sample of N=57 by gender or age distribution because of concerns with statistical power. Of the 57 participants, a large proportion (70%) are males, attributable to the large number of individuals in our sample (e.g., 44%) had a diagnosis of autism spectrum disorder. None of the most frequently used standardized measures of adaptive behavior (e.g., VABS, ABAS, DABS) have reported a sex/gender effect on adaptive functioning. Males and females have typically presented comparable levels of adaptive behavior. We have the same concern regarding the size of sample regarding the running a regression type analysis comparing IQ-adaptive behavior against age. We have added language in our discussion highlighting this limitation of our sample size – limiting our ability to examine if there are any moderating effects on the size of the correlation coefficient of these two constructs.

  1. The major weakness of the article is its sample size. 57 participants is too small to have statistically significant and robust results. It is not clear from the article whether any previous studies were conducted to determine the sample size. There are formulas for doing so, and even packages in different statistical software.

Response: Yes. We inserted a citation in the results section of a statistical reference published in a peer-reviewed journal [VanVoorhis, C. W., & Morgan, B. L. (2007). Understanding power and rules of thumb for determining sample sizes. Tutorials in quantitative methods for psychology3(2), 43-50.] that states that N>50 yields sufficient statistical power when using correlation coefficients to estimate the relationship between two variables.

  1. The paper is very well written and the literature is well analysed. However, there are some points I would like to know about this study that I think the authors should clarify: The sample is distributed between children and adults, but nothing is said about the age or gender distribution of the sample. It would be interesting to see if there are significant differences in the ratios by age or sex. On the other hand, I am concerned about the sample size. I would like you to consider addressing the potential limitations given the small sample size.

Response: Agreed. See response above to comment #3. We have added text to the conclusion section highlighting the limitation of our sample size.

  1. Reviewer 1: The authors should enlarge it or verify that the results they obtain have statistical validity, i.e. they should test the validity of the sample. A typical suggestion is to increase the sample size, but if this is not possible, the authors should use measures to ensure the statistical validity of the sample size. What the authors should do: 1. Prove that the sample size guarantees the results obtained, or, failing that, extend it, either with new data or, if possible, with some bootstrap-type simulation (justifying its use).

Response: See our response to Comment # 3. We feel that we had a valid sample size to conduct our limited Pearson correlations to study the relationship between these variables. We limited our discussion to these correlations and mentioned the limitations, nonetheless, of the size of our sample in conducting additional analyses. Our findings are in the end reasonable and consistent with previously published findings studying other measures of adaptive behavior and intelligence.

  1. Reviewer 1: As a minor suggestion: 2.1 See if there are significant differences in the relationship studied between age ranges, as it is highly variable (from 4 to 21 years).

Response: See response to Comments #1 and #2.

  1. Reviewer 1: 2.2 To see if there are significant differences in the studied relationship between genders.

Response: See response to Comment #2. 

Reviewer 2 Report

This is a fairly interesting take on an old issue of IQ and ADL skills. Indeed in today's more intricate complex, computerized, sophisticated world ADL skills ( they did not mention pre--vocational ) may indeed be more important than an IQ score.

There are several small issues that I would bring up-just for discussions sake-- one never knows about the quality of parenting and how much teaching of ADL skills was contributed by the parents and also the schools.

The authors acknowledge the low-moderate correlations-  and that is good- so this is not an earth shattering study by any means-we have been aware of ADL skills, as the authors say --for decades.

A more relevant issue is how to teach ADL skills more effectively and efficiently and deciding on a curriculum for students with mild, moderate, severe and profound intellectual deficits.

Author Response

Manuscript ID: behavsci-2233109
Type of manuscript: Article
Title: Examining the Relationship Between Adaptive Behavior and Intelligence
Authors: Marc J. Tassé *, Minje Kim
Reviewer: 2

Thank you for your review and comments on our aforementioned manuscript.  We are excited at the possibility of publishing our manuscript in Behavioral Sciences.  We have reviewed your comments and suggested edits to our manuscript and we have made edits to our manuscript to address your comments. We believe these edits, resulting from your review and comments, have strengthened our manuscript. Below is a listing of our responses to Reviewer 2 comments.

  1. This is a fairly interesting take on an old issue of IQ and ADL skills. Indeed in today's more intricate complex, computerized, sophisticated world ADL skills (they did not mention pre--vocational) may indeed be more important than an IQ score.

Response: We added some content to our introduction section related to this point regarding the increasing complexities of adaptive behavior and the need to periodically revise our standardized measures of adaptive functioning to capture these growing complexities. This also reinforces the relevance of conducting our study of adaptive behavior as measured on one of the most recently standardized scales and verifying the correlation between these results and the person’s measured intelligence. See Introduction section, lines 110-115.

  1. There are several small issues that I would bring up-just for discussions sake-- one never knows about the quality of parenting and how much teaching of ADL skills was contributed by the parents and also the schools.

Response: Excellent points about teaching of adaptive behavior skills in the several paragraphs of our discussion section. Also, we added some text in the discussion section highlighting previously published findings examining the effects of parental expectations and adaptive behavior in relation to adult outcomes. These potential factors have been included in our discussion.

  1. The authors acknowledge the low-moderate correlations - and that is good - so this is not an earth shattering study by any means-we have been aware of ADL skills, as the authors say --for decades. A more relevant issue is how to teach ADL skills more effectively and efficiently and deciding on a curriculum for students with mild, moderate, severe and profound intellectual deficits.

Response: See response above and added content to our discussion section, lines 257-263.

Author Response

Manuscript ID: behavsci-2233109
Type of manuscript: Article
Title: Examining the Relationship Between Adaptive Behavior and Intelligence
Authors: Marc J. Tassé *, Minje Kim
Reviewer: 3

Thank you for your review and comments on our aforementioned manuscript.  We are excited at the possibility of publishing our manuscript in Behavioral Sciences.  We have reviewed your comments and suggested edits to our manuscript and we have made edits to our manuscript to address your comments. We believe these edits, resulting from your review and comments, have strengthened our manuscript. Below is a listing of our responses to Reviewer 3 comments.

  1. The main issues for clarification pertains to your focus in the abstract and the methods or what you are working to report here in this study. In the abstract you state using the DABS and a sample of youth with and without ID you examine the correlation coefficients outline. In reading the abstract, the expectation was that you were doing a comparison of the results between the cohorts. That is a report on the strength of the correlational relationship between adaptive skills and IQ Total scores for youth with and then for youth without ID. You work to define these two cohorts in your Participants section stating 51% of participants had intellectual disability (n=29) with the remaining participants having a diagnosis of ASD (n=28). However, you then lump the results together in Table 1 which seems inconsistent with your focus. It would be better to separate the two groups and report on the outcomes for each.

Response: Good observation.  We have rewritten our abstract to reduce the ambiguity in the text that may lead the reader to believe we are comparing two groups of individuals: (1) with ID and (2) without ID. In fact, because we wish to maintain the entire sample size to achieve sufficient statistical power in our correlational analyses, it is best to keep the sample intact and analyze our data with the combined sample of N=57, or option “1” that you list below.

  1. There is further confusion in the methods sections, as you state you looked at 57 participants (which with) studies findings of levels of correlational relationships for individuals with ID. If this was your goal, then you should remove any mention of those without an ID from your abstract and throughout the manuscript.

Response: Done. We removed the references to “with and without ID” throughout.

  1. This leaves you with two choices for focus:
    1. Are you looking at examining the strength of the correlational relationship between components of IQ score and adaptive behaviour ratings with the diagnoses of ID using the DABs to see if there is any difference between previous scales and the DABS? That is, does an updated scale of adaptive behaviour result in a stronger correlation or the same correlation as previous scales for individuals with ID, or….
    2. Are you looking at examining the strength of the correlational relationship between components of IQ score and adaptive behaviour ratings with the diagnoses of ID using the DABs by comparing outcomes of those with a diagnosis of ID and those with a diagnosis of ASD only? That is, you would expect to see a lower correlation between adaptive skills and IQ for those diagnosed with ASD only than those diagnosed with ID or if they are the same, then it adds weight to the argument that the relationship is not causal but rather correlational, given individuals with ASD do not have a diagnosis of ID. I believe your study has a contribution to make with either focus but you should pick one and be clear throughout what you have examined, why and how it contributes. A singular focus will also help strengthen your discussion area. This should be the focus of your revisions.

Response: We think option “a” is best and allows us sufficient statistical power to answer our research question: What is the relationship between adaptive behavior and intelligence?

I have noted a number of minor editing issues, suggestions and areas for clarification as per below:

  1. L30 limitations rather than limitation

Response: Corrected.

  1. L44 Traditionally the DSM version is represented in roman numerals: DSM-V. I would recommend amending throughout.

Response: Historically, the American Psychiatric Association use Roman numerals to designate the different editions of the DSM (DSM-II through DSM-IV/DSM-IV-TR); However, that changed with the publication of the 5th edition. The APA chose to use Arabic number “5” rather than the Roman numeral “V”. For this reason, we kept it as “DSM-5”.

  1. L47 put forth by the APA

Response: Corrected.

  1. L64-65 be consistent in your use of acronyms. Check throughout that you have introduce the full name of the test and then introduces the acronym or abbreviation.

Response: Done.

  1. L127 It is vital or crucial rather than re-use important in the same line.

Response: Replaced “important” with “crucial” – see L115.

  1. L127 as it is a new paragraph restate the constructs for greater clarity: these two important constructs of IQ score and adaptive behaviour….

Response: Done – see L116.

  1. L153 with the reporting of the 44% diagnosed with an autism spectrum disorder (ASD), was this a co-morbid diagnosis or were you specifically looking at this cohort as your “without intellectual impairment group? This is important as ASD no longer includes intellectual disability as a component. If those participants were diagnosed with ASD then be clear that they are your “without” cohort. This would also be a factor to consider in limitations as ASD can impact on adaptive behaviour skills in and of itself. Though you mention in your abstract that you looked at those with and without intellectual disability, there is no comparison data which would have been a good analysis to undertake given the statement in your abstract.

Response:
We cleaned up the language around “with” and “without” ID throughout our manuscript. Previously published research from Alexander and Reynolds (2020) conducted a very large meta-analysis of research examining the relationship between adaptive behavior and intelligence and found that diagnosis of ASD did not impact the size of the relationship when comparing people with ASD and/or ID.  We inserted a statement to this effect in the discussion section, see L234-236.

  1. L165 with have a restricted range: I assume the word with needs to be deleted

Response: Deleted.

  1. L167 the word reduced is repeated and one instance needs to be deleted

Response: Deleted duplicate word.

  1. L127 insert “a” before de-identified dataset

Response: Phrase has been corrected.

  1. L153 you should state the number of participants for each group not just one, so (n=28) for ASD diagnosis cohort

Response: Corrected.

  1. L178 The first sentence needs to be reviewed for clarity. Why did you only include those with intellectual disability when you state in your abstract you looked at those with and without ID?

Response: Corrected.

  1. L185 ae should be are

Response: Corrected.

  1. L187 replace “and” with “to” provide….

Response: Corrected.

  1. L218 restate for clarity, did you mean it was the domain that correlated the highest with the full-scale IQ score and the correlation was even higher than with the DABS Total Score?

Response: Corrected, see L214-216.

  1. L226-227 To strengthen this claim, you could contextualise that even with revised adaptive skills tests such as the DABS the results of this study confirm the ongoing importance of….

Response:
Done.

  1. L229-240 You are restating what you noted in the introduction and given it was repealed has little bearing here.

Response: Done.

  1. L252 change our to the knowledge base L263-268 This section should come earlier and should be part of the impetus and importance for the study.

Response: Moved sentence to beginning of Discussion section.

Round 2

Reviewer 1 Report

The authors have done a great job answering all my questions and doubts. Too bad they can't do anything about the gender issues raised.
Good job!

Reviewer 3 Report

I appreciate the amendments you have made to this paper for greater clarity for a readership which were thoughtful and well addressed.